# Cardiac Arrhythmias in Post-COVID Syndrome: Prevalence, Pathology, Diagnosis, and Treatment

**DOI:** 10.3390/v15020389

**Published:** 2023-01-29

**Authors:** Aydin Huseynov, Ibrahim Akin, Daniel Duerschmied, Rüdiger E. Scharf

**Affiliations:** 1Department of Medicine, Cardiology, Angiology, Hemostasis, and Intensive Care Medicine, University Medical Center Mannheim, Faculty of Medicine Mannheim, University of Heidelberg, DE 68167 Mannheim, Germany; 2Division of Experimental and Clinical Hemostasis, Hemotherapy, and Transfusion Medicine, and Hemophilia Comprehensive Care Center, Institute of Transplantation Diagnostics and Cell Therapy, Heinrich Heine University Medical Center, DE 40225 Düsseldorf, Germany

**Keywords:** post-COVID syndrome, cardiac arrhythmias and channelopathies, myocardial injury, systemic inflammation, cytokines

## Abstract

An increase in post-COVID patients with late sequelae of acute COVID-19 infection is emerging as an ongoing challenge for physicians and healthcare professionals. Since the beginning of the pandemic, it has rapidly become evident that the acute infection is not limited to the respiratory tract but that several organs, including the cardiovascular system, can be affected. Moreover, in a significant proportion of patients (ranging from about 10 to up to 50%) with former COVID-19, cardiopulmonary symptoms such as dyspnea, palpitations, restricted physical capacity, and cardiac arrhythmias can persist weeks and months after the acute SARS-CoV-2 infection. The spectrum of COVID-19-associated arrhythmias is rather wide, most likely due to various pathomechanisms. In this article, the prevalence of cardiac arrhythmias and underlying pathologies are reviewed, including direct myocardial injury and abnormal consequences with an impact on cardiac electric instability. The hyperinflammatory reaction of the host immune system is specifically considered. Moreover, several distinct rhythm disorders occurring in post-COVID patients are discussed with regard to their clinical management.

## 1. Introduction

The impact of the coronavirus pandemic on humanity is undoubtedly one of the worst catastrophes in recent decades. As the pandemic progressed, there was increasing evidence that some patients developed persistent symptoms, often affecting multiple organ systems beyond the acute COVID-19 infection [1,2]. In most cases, the infection causes mild cold symptoms such as fever, cough, and tiredness. However, a significant proportion of patients develop severe symptoms leading to pulmonary injury and ARDS, as well as multiple organ failure with a lethal outcome [3,4]. Persistent fatigue, a change in or loss of smell, palpitations, and chest pain are common symptoms that patients complain about weeks or even months after having experienced an acute COVID-19 infection [5]. This condition is referred to as post-COVID syndrome. It is defined by persistent symptoms that last for 4 to 12 weeks or even several months following an acute COVID-19 infection [6]. The prevalence of post COVID syndrome is not easily ascertained and varies depending on how the time frame is set. The British National Institute for Health and Care Excellence (NICE) guidelines define post-COVID syndrome as the persistence of symptoms beyond 4 weeks of SARS-CoV-2 infection [7]. The World Health Organization describes the post-COVID condition as the 3-month persistence of symptoms lasting at least 2 months and not explained by another illness. Some patients experience the new onset of symptoms after initial recovery from an acute COVID-19 infection, while others display a persistent illness [8]. In a significant proportion of patients (ranging from about 10 to 50%), cardiopulmonary symptoms such as dyspnea, palpitations, restricted physical capacity, and cardiac arrhythmias can persist for weeks and months after acute SARS-CoV-2 infection [8,9]. The spectrum of COVID-19-associated arrhythmias is rather wide, most likely due to various pathomechanisms.

This article reviews cardiac arrhythmias as a distinct feature of post-COVID syndrome. Their prevalence, underlying pathomechanisms, clinical features, diagnosis, and treatment are discussed.

## 2. Epidemiology

Since the beginning of the pandemic, cardiovascular disorders have been reported as the most common extrapulmonary manifestation of SARS-CoV-2 infection [4]. It is estimated that approximately 20 to 30% of hospitalized COVID-19 patients develop cardiac complications. This condition is associated with increased mortality [5,10]. Cardiac involvement in acute COVID-19 has a variety of patterns, ranging from asymptomatic elevations in cardiac biomarkers to cardiogenic shock and sudden cardiac death. Several acute disorders, such as myocardial infarction and myocarditis, which can lead to cardiomyopathy, brady- or tachyarrhythmias, myogenic or arrhythmogenic heart failure, and cardiogenic shock, can occur during the acute COVID-19 infection [11,12].

There is now increasing evidence that these disorders may provide a basis for subsequent cardiac arrhythmias among patients who have experienced a SARS-CoV-2 infection [13]. It is documented that post-COVID symptoms occur more frequently in patients with a severe course of the SARS-CoV-2 infection. In particular, cardiac complaints (e.g., chest pain, shortness of breath, palpitations, and “unspecific” fatigue) can develop after the acute illness and last for months [14]. Several large studies have reported on persistent cardiac arrhythmias after SARS-CoV-2 infection [15]. The overall prevalence of cardiac arrhythmias ranges from 10 to 20% [16]. The numbers and rates indicated above and below may be different with regard to more recent SARS-CoV-2 variants and also vary country-specifically. Notably, the incidence is higher among individuals with severe COVID-19 infection [17]. Persistent cardiac arrhythmias can result from different pathologies, such as primary cardiac injury, secondary cardiac involvement, or impairment of previous cardiovascular disease [13].

## 3. Prevalence and Symptoms of Cardiac Arrhythmias in Post-COVID

While fatigue (97.7%) and intermittent headaches (91.2%) are the most commonly reported symptoms in post-COVID-19 patients, cardiopulmonary symptoms such as chest pain, shortness of breath, and tachycardia are also frequent, affecting approximately one in ten post-COVID-19 patients [18]. Among cardiovascular complaints, tachycardia and palpitations, as well as the subjective perception of cardiac arrhythmia, are the most frequent complaints and are reported by about two-thirds of patients. By contrast to tachycardia, bradyarrhythmia is observed less frequently and documented in only one-third of affected post-COVID-19 patients [19].

Apart from incidentally detected findings by routine ECG, the nature and type of the possible heart rhythm disorder differ depending on the accuracy and design of a questionnaire. In most cases, this is a subjective assessment of the well-being of the people surveyed.

In a 6-month follow-up cohort study on more than 1700 discharged patients with previously proven COVID-19, 154 of 1655 individuals (9.3%) suffered from palpitations [5]. Davies et al. conducted an online survey on a population with suspected or confirmed COVID-19 at 6 months after acute illness and analyzed responses from 3762 participants [19]. Of the 2308 patients (61.4%) who indicated tachycardia, 1680 had measured their heart in standing vs. sitting position. Of those, 515 patients (30.7%) displayed an increase in heart rate of at least 30 BPM when in an upright position. As a conceivable explanation, the authors assumed postural orthostatic tachycardia syndrome (POTS) in this subgroup of responders [19]. However, the nature and extent to which the subjective complaints can point to specific cardiac arrhythmias remain open. In fact, most observational post-COVID studies are based on general complaint questionnaires and not designed to assess specific entities or disorders such as cardiac arrhythmias.

## 4. Risk Factors and Associated Conditions

Despite the great variability regarding the kind and frequency of individual complaints in published studies, the panel of risk factors and associated conditions for heart-specific complaints are similar in most studies (Figure 1). Female gender, age over 70 years, diabetes, obesity, comorbidity, and the need for inpatient treatment of COVID-19 were shown to be associated with an increased risk of suffering from post-COVID symptoms [18,20]. Among these risk determinants, obesity appears to play a distinct role. It is reported that being overweight increases the likelihood of post-COVID by about 25% [21]. This association can be explained by the fact that obesity is often related to the other risk determinants indicated above.

Sociodemographic characteristics (older age, female gender, white ethnicity) and concomitant variables (e.g., poor mental and physical health, asthma) are being considered as additional risk factors in adult patients for developing post-COVID syndrome. This conclusion is also derived from the analysis of survey data obtained from 6907 of 48,901 individuals (14.1%) with self-reported COVID-19, recorded by 10 UK longitudinal study samples, and from 1.1 million individuals with COVID-19 diagnostic codes, collected in electronic healthcare records [21]. The rate of presumed COVID-19 cases reporting persistent symptoms for more than 12 weeks after initial illness ranged from 7.8% to 17%. Furthermore, increasing age, female gender, white ethnicity, poor pre-pandemic physical and mental health, overweight/obesity, and asthma were associated with prolonged symptoms; however, findings for other variables and determinants, such as cardiological and metabolic parameters, were inconclusive [21].

Apart from the SARS-CoV-2 infection, comorbidity, and physical factors, the social situation may also have an impact on the persistency of post-COVID symptoms [22,23,24]. For example, numerous lockdowns, unemployment, reduced physical activity, multiple social fears, anxiety, depression, and economic challenges due to the COVID-19 pandemic can also trigger various cardiac complaints [8].

## 5. Pathogenesis of Cardiac Arrhythmias in COVID-19

The unexpectedly high prevalence of cardiac arrhythmias, ranging from 10 to 20% after COVID-19, requires careful exploration of the pathology leading to myocardial injury and, subsequently, an increase in myocardial electric instability.

Overall, the spectrum of cardiac arrhythmias after SARS-CoV-2 infection is wide, comprising supraventricular tachycardias, in particular atrial fibrillation, as well as ventricular tachyarrhythmias, bradyarrhythmias, and conduction defects (Figure 2). However, the underlying mechanisms of cardiac arrhythmias during and after COVID-19 are still ill-defined [8].

## 6. Direct Cellular Injury

The host cell receptor for SARS-CoV-2 is the biologically critical enzyme angiotensin-converting enzyme (ACE2). This receptor is expressed in tissues of the heart, lung, kidney, nose, mouth, and pharynx cavity [25], while nasal, oropharyngeal, and pulmonary epithelial cells are identified as primarily SARS-CoV-2-infected cells upon entry, replication, and release into the circulation. Based on gene expression studies, the human ventricular myocardium also contains the requisite mediators of SARS-CoV-2 binding and entry [26]. Interestingly, ACE2 receptors are expressed in myocytes and pericytes but not in endothelial cells [27,28].

Using a rabbit in vivo model of coronavirus-induced myocarditis that was established over 30 years ago, direct viral infection of the heart, leading to myocarditis and subsequent congestive heart failure, had been demonstrated [29,30]. Another model with engineered heart tissue containing human pluripotent stem cell-derived cardiomyocytes, fibroblasts, and macrophages was developed to explore virus-induced direct cellular injury [31]. By contrast, autopsy studies did not prove the hypothesis of direct myocardial lesions in COVID-19. Thus, in most cases, viral RNA was not detectable in cardiomyocytes [32]. Moreover, in a multicenter COVID-19 autopsy series, instead of typical myocarditis features, interstitial macrophage infiltration without cardiomyocyte degeneration was found in over 80% of cases [33].

An alternative hypothesis under investigation is that the renin-angiotensin system (RAS) may be involved in the pathology of COVID-19 via ACE-2 receptor activation. The ACE2 receptor serves as a master regulator of RAS. Beyond causing direct cellular damage through viral infiltration, SARS-CoV-2 downregulates ACE2 expression following unfavorable effects, including vasoconstriction, dysfunction of the endothelium, inflammatory responses (hyper-inflammation), activation of coagulation, and ultimately increased vascular permeability [34].

## 7. Microvascular Injury

A current hypothesis to explain cardiac lesions caused by SARS-CoV-2 infection suggests that the heart tissue is not damaged by direct viral detrimental effects but by hyperactivation of the immune system. This hyperactivation appears to be caused by a misdirected response of the immune-inflammatory system, which in turn leads to microvascular injury and lesions of cardiomyocytes with subsequent arrhythmogenic effects [15]. An exuberant and faulty immune response is currently thought to be the basis of COVID-19-associated cardiac involvement and resulting disorders [35].

Several subsequent mechanisms, such as hypoxia-induced myocardial injury due to hypoxic respiratory failure, small vessel ischemia due to microvascular injury with or without thrombotic occlusion, and acute right ventricular failure due to pulmonary embolism or in situ pulmonary artery thrombosis, are discussed [32]. These pathologies are considered to be the result of indirect myocardial effects of severe SARS-CoV-2 infection. In clinical reality, these pathologies cannot be differentiated from each other, as, in the majority of individual cases, they occur in combination.

Venous or arterial thrombosis and platelet activation play a relevant role in cardiac involvement based on the inflammation of the vascular endothelium (endotheliitis). The endothelial pathways are often complex, with different mechanisms of action leading to enhanced secretion of von Willebrand factor from the Weibel-Palade bodies, its endothelial storage organelles, or resulting in the formation of platelet aggregates and/or platelet-leukocyte complexes in the circulation [36].

A relevant pathology in SARS-CoV-2 infection in general but also in the initiation and progression of myocardial injury is due to the formation of neutrophil extracellular traps (NETs) [36,37]. This mechanism (also designated NETosis) is physiologically part of the host’s defense by trapping invaded pathogens, but under abnormal conditions NETosis is involved in a variety of disorders, including SARS-CoV-2 infection [38,39]. Specifically, the formation of NETs can promote arterial and venous thrombosis [40].

## 8. Inflammatory Cytokines in COVID-19-Associated Cardiac Arrhythmias

By contrast to the initial contention that COVID-19-associated arrhythmias may directly result from disease-specific mechanisms (e.g., myocardial injury due to viral infection and replication with subsequent electric instability), several lines of evidence now suggest that the high-grade inflammatory state that is induced in response to SARS-CoV-2 infection plays an eminent role in the pathogenesis of cardiac arrhythmias [15]. Therefore, it has become apparent that, apart from tissue hypoxia and cardiac injury, as reflected by increased troponin levels, systemic inflammatory activation is a major arrhythmogenic trigger [41]. Interestingly, a direct correlation between elevated troponin levels and markedly increased proinflammatory cytokines was demonstrated in patients with COVID-19 [42].

In general, inflammatory cytokines can have arrhythmogenic effects. This is specifically true for tumor necrosis factor-α (TNF-α), interleukin-1 (IL-1), and IL-6, all of which may induce cardiac arrhythmias [43]. It has also been shown that several mechanisms are involved, including direct cardiac changes and indirect systemic alterations, by which these cytokines can act as drivers to exert relevant arrhythmogenic damage.

## 9. Direct Arrhythmogenic Effects of Inflammatory Cytokines

Direct effects of inflammatory cytokines result from cardiac remodeling, as evident by structural and electrical changes. These changes are characterized by modulatory activities on the expression and function of distinct cardiomyocyte proteins, including cardiac ion channels or gap junction-forming transmembrane proteins (in particular connexins), and specific receptors of calcium ion metabolism or transport [44]. Eventually, the structural and functional alterations result in inflammatory cardiac channelopathies. Of note, electrical changes caused by such channelopathies can occur early (i.e., within hours to days after exposure to cytokines) [15]. Moreover, at later stages (weeks or months), inflammatory cytokines can also induce structural remodeling by activating myoblast-driven synthesis of extracellular matrix proteins, which, in turn, contribute to cardiac fibrosis and subsequent electric disorders [43].

Taken together, structural and functional cardiac changes induced by inflammatory cytokines in response to SARS-CoV-2 infection can lead to cardiac arrhythmias, specifically a prolonged action potential duration or corrected QT interval, enhanced ectopic firing, and/or abnormal propagation of electric impulses throughout the conducting and working myocardium [15]. As displayed in Figure 2, these changes can promote triggered and/or re-entry-driven tachy- and bradyarrhythmias or result in atrial or ventricular fibrillation and atrioventricular blocks [43].

## 10. Indirect Arrhythmogenic Effects

Indirect arrhythmogenic effects in association with COVID-19 (and other infections) result from a variety of systemic disturbances, including *(i)* inhibition of the hepatic cytochrome P450 system (subsequently increasing the bioavailability of concomitant QT-prolonging agents); *(ii)* stimulation of aromatase activity in adipose tissue and enhanced androgen-to-estrogen conversion (promoting action potentials and QT prolongation due to reduced testosterone levels); and *(iii)* several effects on the nervous system, caused by fever and temperature-related changes in biophysical and functional properties of cardiac ion channels [15]. Some of these changes lead to intracellular Ca^2+^ overload, thereby promoting action potential prolongation.

## 11. Potential Mechanisms of Arrythmias in Post-COVID

Similar to other systemic viral disorders such as acute respiratory distress syndrome (ARDS) in influenza, any pre-existing chronic cardiovascular diseases can be impaired by COVID-19-associated hypoxemia. Patients with heart failure (HF) have limited cardiac and respiratory capacity and reserve and, thus, insufficiently tolerate infection-induced cardiac complications [45].

Since a direct cytotoxic effect of SARS-CoV-2 on cardiac myocytes is rather unlikely, unfavorable consequences resulting from the hyperinflammatory response of the innate immune system appear to play a major role. Thus, it has been demonstrated that IL-6, TNF-α, and IL-1 can prolong the ventricular action potential duration by modulating the expression or function of several cardiomyocyte ion channels, specifically K^+^ and Ca^2+^ channels. These changes are referred to as inflammatory cardiac channelopathies. In particular, there is evidence that IL-6 can directly inhibit K^+^ channels and prolong action potential duration in ventricular myocytes [46,47]. Several growth factors, such as transforming growth factor-β (TGF-β), platelet-derived growth factors (PDGF), cytokines (TNF-α, IL-1, IL-6, IL-10, and IL-4), and neurohumoral pathways, can be drivers of myocardial fibrosis by inducing signaling cascades through binding to surface receptors, thereby activating downstream signaling cascades [48,49]. Specifically, the cytokine storm, as it may occur typically in response to SARS-CoV-2 infection associated with severe courses, can trigger structural remodeling due to fibrosis pathway activation [48,49].

In a small study of 26 recovered patients with ongoing cardiac symptoms, cardiovascular magnetic resonance (CMR) imaging revealed cardiac involvement in 58% of patients consisting of myocardial oedema and scar by late gadolinium enhancement (LGE) [50]. Another CMR study reported on cardiac involvement in 78% of patients (including ongoing myocardial inflammation in 60% of them). These findings were made independent of pre-existing conditions, the severity and overall course of the acute illness, and the time since the COVID-19 infection. Peri-epicardial LGE in combination with pericardial effusion can be attributed to fibrosis or oedema due to ongoing active pericarditis [51]. Fibrosis may perturb the propagation of the electrical impulse and generate re-entry circuits, thus contributing to the pathogenesis of arrhythmias. Links between fibrosis and increased risk of arrhythmias have been documented in almost every cardiac pathologic condition using CMR for the assessment of fibrotic remodeling [52,53].

Another possible mechanism for cardiac damage in post-COVID patients is an autoimmune response to cardiac antigens due to molecular mimicry. Anti-cardiac antibodies can be considered as part of the systemic immune and inflammatory response to SARS-CoV-2 [54]. Patients with COVID-19 also exhibit marked increases in autoantibody reactivities as compared with uninfected individuals and display a high prevalence of autoantibodies triggering chronic inflammation of the myocardium in post-COVID patients [55,56].

These mechanisms lead to long-term cardiovascular changes after COVID-19 infection and can provide the basis for cardiac arrhythmias in post-COVID patients [51].

## 12. Postural Orthostatic Tachycardia Syndrome (POTS)

POTS and other common autonomic disorders can follow COVID-19 in previously healthy, non-hospitalized patients who experience significant disability after the acute infection [57]. POTS is characterized by an excessive increase in heart rate in an upright position, symptoms of orthostatic intolerance, and occasional syncope. Although postural tachycardia is the leading symptom, it very often occurs with vegetative signs such as general deconditioning, chronic fatigue, exhaustion, heat intolerance, fever, and debility [58].

The syndrome affects younger individuals and has a distinct predominance among women (about 80%). The onset of POTS may be triggered by typical immunological stressors such as viral infection, vaccination, trauma, pregnancy, surgery, or even intensive psychological stress [58]. Prior to the COVID-19 pandemic, an antecedent history of suspected viral infection was reported in a large proportion of POTS patients, ranging from 20% to 50% [58,59,60]. Recent studies suggest that patients with POTS have a higher prevalence of autoimmune markers and co-morbid autoimmune disorders than the general population [61]. In about of 20% of POTS patients, elevated levels of autoantibodies or inflammation markers are present, supporting the hypothesis of a persistent general inflammatory reaction involving the cardiovascular system [57]. The symptoms of POTS, especially tachycardia, can be evoked by direct effects on the sinus rate-controlling system (via adrenergic and muscarinic receptors) or through a compensatory mechanism responding to peripheral vasodilation (via adrenergic, angiotensin, and possibly other vasoactive receptors). These abnormalities suggest a direct association of a former COVID-19 infection with the evolution of chronic inflammation and the development of POTS due to persistent autoantibodies.

Both the heterogeneity and the wide spectrum of POTS symptoms present a major challenge to diagnosing and managing this disorder. Randomized controlled studies on post-COVID-associated POTS therapy are not available. Thus, any treatment of POTS is not evidence-based.

Therapeutic options include non-pharmacological and pharmacological approaches. General measures, moderate physical exercise, and/or fluid and salt intake are applied to prevent orthostatic hypotension [62,63,64]. Pharmacologically, the use of midodrine, ß-blockers, fludrocortisone, pyridostigmine, clonidine, or alpha-methyldopa is recommended. Overall, the treatment of POTS can be difficult. In fact, there are no therapies that are uniformly successful, and combinations of approaches are often applied [65].

## 13. Inappropriate Sinus Tachycardia Syndrome (IST)

IST, a rather common observation in post-COVID-19 patients, is defined by a sinus heart rate of >100 bpm at rest (with an unexplained mean 24-h heart rate > 90 bpm) and is associated with distressing symptoms of palpitations [65]. In a study on post-COVID patients, approximately 20% of them met the criteria of the IST [66]. The disorder was more common in young women without previous morbidities and a mild SARS-CoV-2 infection.

The phenomenon of IST after viral infections was already known prior to COVID-19. IST is described in up to 40% of patients after suffering from SARS-CoV-2 infection, a similar frequency that is also observed after other viral disorders such as middle respiratory syndrome and diseases caused by the human immunodeficiency virus or the Epstein-Barr virus [67]. IST usually occurs in combination with vegetative symptoms such as fatigue, reduced resilience, and inappropriate dyspnea on exertion [66,67,68]. Similar to the other cardiac arrhythmias in post-COVID patients, the pathology of IST has not yet been explicitly clarified. Various mechanisms, such as direct or indirect damage to the central and peripheral nervous system, can be involved. However, it remains unclear why predominantly female patients are affected by IST and why the severity of the syndrome does not correlate with the severity of the former COVID-19 [66]. Patients with IST may likely benefit from pharmacological treatment analogous to that for POTS using ß-blockers, which blunt the sympathetic nervous system response. However, there are no recommendations for a standardized management of such patients. Therefore, future intervention studies are needed for the optimal management of IST in post-COVID patients [69].

## 14. Atrial Fibrillation (AF) in Post-COVID Patients

AF is one of the most common cardiovascular disorders and shares similar comorbidities with patients suffering from post-COVID syndrome. In general, AF has a high prevalence in populations with advanced age, cardiovascular risk factors, and comorbidities [70]. Several studies have reported on a significantly increased mortality rate among inpatients with atrial fibrillation. Of particular interest is the occurrence of AF during hospitalization. It has been demonstrated that the new onset is an independent predictor of worse outcomes, such as in-hospital mortality, the need for mechanical ventilation, and cardiovascular death [70,71,72,73].

AF occurs in a subset of hospitalized patients with COVID-19 who are suffering from serious disease progression and display hyperinflammatory laboratory features [71]. However, a critical analysis of published studies does not allow a differentiation as to which proportion of the patients had undetected AF prior to hospitalization and in which patients the active COVID-19 infection actually induced the occurrence of the AF. It is also difficult to differentiate in post-COVID patients which subjects suffer from atrial flutter triggered by the acute infection and in whom the post-COVID condition is the true cause.

Post-COVID-induced myocardial lesions associated with AF may be evoked by a variety of pathologies, including hypercoagulability or other hemostatic disorders, including the formation of extracellular neutrophil traps (NETosis), persistent hyperinflammation, and concomitant myocardial injury of other etiologies [14,37,39]. Thus, the presence of AF is correlated with adverse outcomes in patients with COVID-19 pneumonia, which deserves increased attention and should be managed appropriately to prevent adverse outcomes [70,74].

The analysis of COVID patients who did not need hospital care, presumably due to a milder course of the infection, can also be informative and insightful. Thus, there is an increasing number of reports documenting that the prevalence of arrhythmias in the first 6 months after COVID-19 is significantly higher than in the population without infection. For example, a retrospective analysis of the Veterans Health Administration electronic medical record revealed a 1.7 times higher incidence of AF at 6-months after former SARS-CoV-2 infection in non-hospitalized patients compared with matched controls [75]. To date, treatment of the AF in post-COVID patients has been carried out according to the general guidelines for atrial fibrillation. Currently, there are no recommendations that persistent atrial fibrillation should be treated differently in the post-COVID setting.

## 15. Ventricular Arrythmias and Sudden Cardiac Death

The clinical implications of ventricular arrythmias in post-COVID patients are limited by the restricted options for detecting cardiac arrhythmias. Therefore, available data are derived from studies with cardiac device wearers, professional athletes, hobby sportsmen/women, or other population subgroups.

A study from Italy, enrolling 696 patients with cardiac devices, reported on the effects of COVID-19 and demonstrated a significant increase in the burdens of ventricular tachycardia, associated with a decrease in physical activity and heart rate variability [76]. Considering the trend of reduced physical activity in subjects with post-COVID conditions, this phenomenon appears to be associated with a direct impact of the reduced physical activity on ventricular arrythmias [76]. Another multicenter trial studying 204 post-COVID patients in a 3-month follow-up after hospitalization detected mildly impaired right ventricular function and higher rates of ventricular arrhythmias compared with a control group [77]. A higher proportion of cardiac arrhythmias was observed in 27% of post-COVID patients. Premature ventricular contractions (PVC) were the most frequent abnormality recorded in 18% of patients (with a mean of 1300 PVC/day). There were significantly more dangerous non-sustained ventricular tachycardias in 5% of the patients; however, most of them had a history of structural heart disease such as ischemic or dilated cardiomyopathy [77].

With regard to cardiac arrhythmias, observational studies on athletes are of particular interest, as younger individuals without structural heart disease are screened here for the presence of arrhythmias after former COVID-19 diagnoses. Competitive athletes are usually young and healthy, and most of them have an asymptomatic or mild course of the SARS-CoV-2 infection [78]. In general, low rates of cardiac arrhythmias or sudden cardiac death are reported collectively for these individuals [79].

In a prospective observational study enrolling 3018 young athletes (mean age, 20 years) after COVID-19, a low prevalence (0.5% to 3.0%) of definite, probable, or possible SARS-CoV-2-induced myocardial involvement and a low risk of adverse cardiac events were detected. Only one sudden cardiac death (successfully resuscitated sudden cardiac arrest) was reported, but it was most likely unrelated to viral cardiac involvement observed [80]. In a study on 30 male professional soccer players after COVID infection, only one athlete had ventricular premature beats during exercise. In another individual sportsman, supraventricular premature beats (SVPBs) during exercise were detected that had not been present at exercise testing prior to the SARS-CoV-2 infection [81]. Significantly higher rates of isolated ventricular premature beats (VPB) and SVPB were found in 53.3% and 52.5% of 90 post-COVID athletes, respectively, but no malignant arrhythmias were identified [82].

## 16. COVID-19 Vaccination-Associated Arrhythmias

Cardiovascular complications after vaccination against SARS-CoV-2 are of particular interest. In fact, multiple cases of vaccination-associated cardiac disorders have been reported when using mRNA-based vaccines. Myocarditis has been recognized as a rare complication of the COVID-19 mRNA vaccination, specifically among adolescent males and young adults [83,84]. According to the US Centers for Disease Control and Prevention, myocarditis rates are about 12.6 cases per one million doses of second-dose mRNA vaccine among individuals at 12 to 39 years of age [85].

Some cases of cardiac arrhythmias, including non-sustained ventricular tachycardia [86], atrial fibrillation [87], high-degree atrioventricular block [88], have also been described without previous clinical signs of myocarditis; however, these observations were made without definitely ruling out any myocardial injury by using cardiac MRI. A large study enrolling 7934 high school students after mRNA vaccination and ECG screening revealed that the second dose of anti-SARS-CoV-2 vaccination with a mRNA-based vaccine can have significant cardiac adverse effects. Thus, among 4928 eligible study participants, 763 students (17.1%) had at least one cardiac symptom (mostly chest pain and palpitation); abnormal ECG changes were obtained in 51 subjects (1.0%), of whom one was diagnosed with mild myocarditis whereas four were judged to have significant arrhythmia [89]. Similarly, very low rates of post-vaccination arrhythmias (about 0.2–1.0%) were observed in a UK study of over 38 million people who received a COVID-19 vaccine, but relatively higher rates were seen in those who had been tested SARS-CoV-2 positive [90].

By contrast to the general population, the incidence of arrhythmias significantly increases in patients with cardiac implantable electronic devices (CIED) after SARS-CoV-2 vaccination. Thus, a 73% increase in the incidence of supraventricular tachycardias and ventricular arrhythmias was observed among 180 patients with CIED during the post-vaccination period, but no life-threatening arrhythmias were noted [91].

Fever-related arrhythmic events are another relevant aspect of vaccination-induced adverse effects. For example, fever can be associated with an increased rate of life-threatening arrhythmias in subjects with Brugada syndrome [92]. As reported in a study of 163 patients with Brugada syndrome, about 20% of them experienced fever (>37.9° Celcius) after anti-SARS-CoV-2 vaccination; however, due to prompt antipyretic treatment, no new ECG features were recorded among study patients following vaccination [93].

## 17. Conclusions

During the COVID-19 pandemic, millions of people around the world have become infected by the virus. Even though the course of COVID-19 due to more recent SARS-CoV-2 variants is not as dramatic as at the beginning of the pandemic, the healthcare system remains increasingly concerned with the consequences, complications, and costs of COVID-19. As the number of people who have experienced the infection increases, so does the number of individuals who suffer from post-COVID symptoms. Cardiac arrhythmias are some of the most relevant manifestations of this condition and lead to a significant reduction in the quality of life. The spectrum of cardiac arrhythmias in post-COVID patients is wide and variant, probably due to different pathological mechanisms that need further exploration.

The spectrum of possible cardiac arrhythmias is broad, ranging from benign ventricular extrasystoles to atrial fibrillation and sudden cardiac death from ventricular arrhythmias. Thereby, the frequency of occurrence clearly correlates with cardiovascular comorbidities. However, cardiac arrhythmias can also affect previously healthy individuals or athletes. Together with other symptoms of the post-COVID condition, cardiac arrhythmias cause a further reduction in the quality of life.

Future research priorities related to the nature and pathogenesis of arrhythmias will focus on mechanisms of cardiac injury, including the role of a deviant immune system, dysfunctional endothelium, hemostatic changes, and abnormalities of the nervous system. Vaccination and the long-term analysis of its effects represent another important aspect, both in research and clinical management. Overall, a multidisciplinary approach will be required to improve our understanding of the broad spectrum of post-COVID-19 conditions in subjects with prior SARS-CoV-2 infection.

Cardiac arrhythmia screening in symptomatic patients is required to make the correct diagnosis and provide patients with adequate therapy. The nature of testing and the efficiency of screening procedures for post-COVID arrhythmias have yet to be determined. Future studies, both observational and interventional, are needed to develop evidence-based recommendations for post-COVID patients.

## Figures and Tables

**Figure 1 viruses-15-00389-f001:**
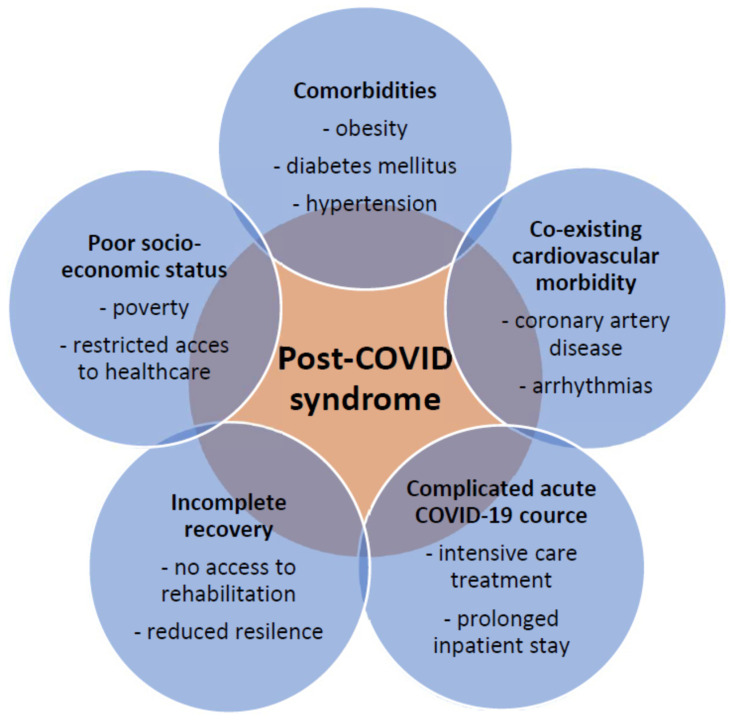
Post-COVID syndrome and conditions that are associated with an unfavorable course.

**Figure 2 viruses-15-00389-f002:**
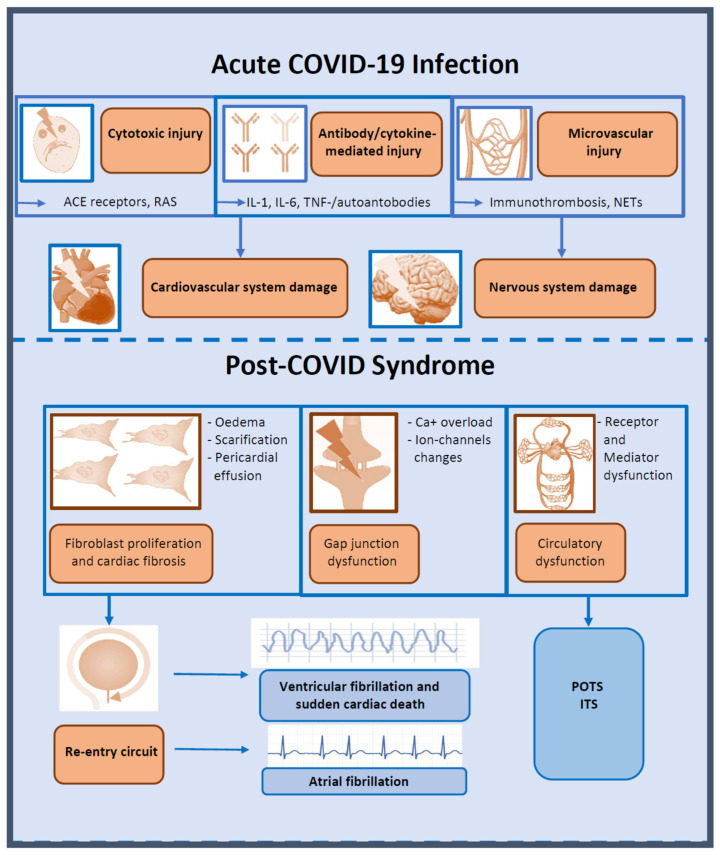
Arrhythmias in post-COVID syndrome: possible pathologies and common arrhythmias during and after COVID-19 infection. ACE, angiotensin-converting enzyme; IL, interleukin; ITS, inappropriate tachycardia syndrome; NETs, neutrophil extracellular traps; POTS, postural tachycardia syndrome; RAS, renin-angiotensin system; TNF, tumor necrosis factor.

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
