# Peer review of "Cardiac Arrhythmias in Post-COVID Syndrome: Prevalence, Pathology, Diagnosis, and Treatment"

_viruses, 2023, doi:10.3390/v15020389_

Round 1
Reviewer 1 Report
The article is very well organized and nearly all the relevant information has been collected and interestingly presented.
I have a few significant changes, which I would like to mention.
1. Introduction is too broad, it should be in the same tone as the topic indicates, cardiac complications should also be discussed here.
2. Conclusion is too short and does not suit the body of the article, which is very well-written and organized.
3. could you add another section highlighting the similar covid-like symptoms and cardiac complications due to vaccines? maybe pseudo-pathological conditions or complications.
Minor.
1. Figure 1 should be converted to anything like SmartArt (Hierarchy type)
2. in the prevalence-epidermiology section, you have stated very strong statements, maybe you need to discuss the variant of COVID here too. maybe country too.
3. in the microvascular injury section, there are fewer references, need to look into this section.
Author Response
Reviewer 1:
- Introduction is too broad, it should be in the same tone as the topic indicates, cardiac complications should also be discussed here.
In accord with the reviewer’s suggestion, the last two sentences of paragraph 1 of the Introduction were deleted, and the following statement was added:
In a significant proportion of patients (ranging from about 10 to up to 50%), cardiopulmonary symptoms such as dyspnea, palpitations, restricted physical capacity, and cardiac arrhythmias can persist for weeks and months after acute SARS-CoV-2 infection. The spectrum of COVID-19-associated arrhythmias is rather wide, most likely due to various pathomechanisms.
- Conclusion is too short and does not suit the body of the article, which is very well-written and organized.
The Conclusion section has been extended and adjusted. The following paragraph has been added:
The spectrum of possible cardiac arrhythmias is broad, ranging from benign ventricular extrasystoles to atrial fibrillation and sudden cardiac death from ventricular arrhythmias. Thereby, the frequency of occurrence clearly correlates with cardiovascular comorbidities. However, cardiac arrhythmias can also affect previously healthy individuals or athletes. Together with other symptoms of the post-COVID condition, cardiac arrhythmias cause a further reduction in the quality of life.
Future research priorities related to the nature and pathogenesis of arrhythmias will focus on mechanisms of cardiac injury, including the role of a deviant immune system, dysfunctional endothelium, hemostatic changes, and abnormalities of the nervous system. Vaccination and long-term analysis of its effects represents another important aspect both in research and clinical management. Overall, a multidisciplinary approach will be required to improve our understanding of the broad spectrum of post-COVID-19 conditions in subjects with prior SARS-CoV-2 infection.
- could you add another section highlighting the similar covid-like symptoms and cardiac complications due to vaccines? maybe pseudo-pathological conditions or complications.
We entirely agree with the reviewer and thank him/her for this important suggestion. Accordingly, we have now added a corresponding section about vaccination-induced cardiac arrhythmias.
COVID-19 vaccination-associated arrhythmias
Cardiovascular complications after vaccination against SARS-CoV-2 are of particular interest. In fact, multiple cases of vaccination-associated cardiac disorders have been reported, when mRNA-based vaccines. Myocarditis has been recognized as a rare complication of COVID-19 mRNA vaccination, specifically among adolescent males and young adults. According to the US Centers of Disease Control and Prevention, myocarditis rates are about 12.6 cases per one million doses of second-dose mRNA vaccine among individuals at 12 to 39 years of age.
Some cases of cardiac arrhythmias, including non-sustained ventricular tachycardia, atrial fibrillation, high-degree atrioventricular block, have also been described without previous clinical signs of myocarditis; however, these observations were made without definitely ruling out any myocardial injury by using cardiac MRI. A large study enrolling 7934 high school students after mRNA vaccination and ECG screening revealed that the second dose of anti-SARS-CoV-2 vaccination with a mRNA-based vaccine can have significant cardiac adverse effects. Thus, among 4928 eligible study participants, 763 students (17.1%) had at least one cardiac symptom (mostly chest pain and palpitation); abnormal ECG changes were obtained in 51 subjects (1.0%), of whom one was diagnosed with mild myocarditis, whereas four were judged to have significant arrhythmia . Similarly, very low rates of post-vaccination arrhythmias (about 0.2-1.0%) were observed in a UK study of over 38 million people who received a COVID-19 vaccine, but relatively higher rates were seen in those who had been tested SARS-CoV-2 positive.
By contrast to the general population, the incidence of arrhythmias significantly increases in patients with cardiac implantable electronic devices (CIED) after SARS-CoV-2 vaccination. Thus, a 73% increase in in incidence of supraventricular tachycardias and ventricular arrhythmias was observed among 180 patients with CIED during the post-vaccination period, but no life-threatening arrhythmias were noted.
Fever-related arrhythmic events are another relevant aspect of vaccination-induced adverse effects. For example, fever can be associated with an increased rate of life-threatening arrhythmias in subjects with Brugada syndrome. As reported in a study of 163 patients with Brugada syndrome, about 20% of them experienced fever (> 37.9° Celcius) after anti-SARS-CoV-2 vaccination; however, due to prompt antipyretic treatment no new ECG features were recorded among study patients following vaccination.
Minor.
Figure 1 should be converted to anything like SmartArt (Hierarchy type)
We have followed the reviewer’s for proposal and used SmartArt for Fig. 1, as displayed in attachment. However, we feel that the original presentation is superior and would therefore prefer to leave it unchanged.
- in the prevalence-epidermiology section, you have stated very strong statements, maybe you need to discuss the variant of COVID here too. maybe country too.
We also thank the reviewer for this comment and agree with his/her suggestion. Consequently, we have added the following statement to the Epidemiology section, 2nd paragraph.
“The numbers and rates indicated above and below may be different with regard to more recent SARS-CoV-2 variants and also vary country-specifically.”
- in the microvascular injury section, there are fewer references, need to look into this section.
We appreciate the reviewerr’s comment and have updated the microvascular injury section accordingly by adding relevant references related to “COVID-19 is an endothelial/vascular disease”:
Siddiqi, H. K., Libby, P. & Ridker, P. M. COVID-19 - A vascular disease. Trends Cardiovasc Med 31, 1-5 (2021). https://doi.org:10.1016/j.tcm.2020.10.005
Libby, P. & Luscher, T. COVID-19 is, in the end, an endothelial disease. Eur Heart J 41, 3038-3044 (2020). https://doi.org:10.1093/eurheartj/ehaa623
Libby, P. The Heart in COVID-19: Primary Target or Secondary Bystander? JACC Basic Transl Sci 5, 537-542 (2020). https://doi.org:10.1016/j.jacbts.2020.04.001
Solimando, A. G., Marziliano, D. & Ribatti, D. SARS-CoV-2 and Endothelial Cells: Vascular Changes, Intussusceptive Microvascular Growth and Novel Therapeutic Windows. Biomedicines 10 (2022). https://doi.org:10.3390/biomedicines10092242

Reviewer 2 Report
The article is well written, it is based on detailed literature overview of this actual topic.
Author Response
We thank the reviewer for his/her comment.
Reviewer 3 Report
The authors have researched a relatively new hot topic of CVD. The review is well structured and set to an audience of academic cardiologists. Since the pandemic many efforts have been spent to further understand the implication of COVID-19 with CVDs. A part of these efforts has investigated Arrhytmias and post-COVID syndrome involvement, highlighting interesting scientific evidence.
The article requires moderate English adjustments and there are common punctuation errors.
My concern regards the introduction as authors who have investigated the same field have not been taken into consideration (35159974, 34390682). These authors have highlighted how post-COVID sindrome has different manifestations but regards more than one system. Moreover, the authors talk about a CVD that requires management and a therapeutic consideration is missing. Lastly, the Conclusion is missing a future prospective piece where the authors suggest what future research should embrace.
Author Response
The article requires (1) moderate English adjustments and there are (2) common punctuation errors.
We thank the reviewer for his/her comment and the suggestions. Acorrdingly, we have performed the required adjustments so far as evident for us.
My concern regards the introduction as authors who have investigated the same field have not been taken into consideration (35159974, 34390682).
We also thank the reviewer for this comment referring to the work of other investigators (“authors”). The first reference was added to the introduction and the second one was already added to the inappropriate sinus tachycardia syndrome section.
These authors have highlighted how post-COVID sindrome has different manifestations but regards more than one system. Moreover, the authors talk about a CVD that requires management and a therapeutic consideration is missing.
We thank the reviewer for this relevant comment. Indeed, the management of specific cardiac arrhythmias can be challenging. We have indicated therapeutic considerations or recommendations that are outlined at the end of each section.
Lastly, the Conclusion is missing a future prospective piece where the authors suggest what future research should embrace.
We agree with the reviewer and appreciate his/her comment. Accordingly, we now have extended and specified the Conclusion by adding:
The spectrum of possible cardiac arrhythmias is broad, ranging from benign ventricular extrasystoles to atrial fibrillation and sudden cardiac death from ventricular arrhythmias. Thereby, the frequency of occurrence clearly correlates with cardiovascular comorbidities. However, cardiac arrhythmias can also affect previously healthy individuals or athletes. Together with other symptoms of the post-COVID condition, cardiac arrhythmias cause a further reduction in the quality of life.
Future research priorities related to the nature and pathogenesis of arrhythmias will focus on mechanisms of cardiac injury, including the role of a deviant immune system, dysfunctional endothelium, hemostatic changes, and abnormalities of the nervous system. Vaccination and long-term analysis of its effects represents another important aspect both in research and clinical management. Overall, a multidisciplinary approach will be required to improve our understanding of the broad spectrum of post-COVID-19 conditions in subjects with prior SARS-CoV-2 infection.
